# Human resistin is critical to activation of the NLRP3 inflammasome in macrophages

Udeshika Kariyawasam[1], Winson Lam [1], John Skinner[1], Rituparna Chakrabarti [1], Andrea Cox[2], Paul M. Hassoun [2], Qing Lin [1], Roger A. Johns [1,2]*

1 Department of Anesthesiology and Critical Care Medicine, Johns Hopkins University School of Medicine, Baltimore, Maryland, United States of America, 2 Department of Internal Medicine, Johns Hopkins University School of Medicine, Baltimore, Maryland, United States of America

* rajohns@jhmi.edu

## Abstract

Elevated levels of human resistin (hResistin) have been associated with diverse inflammatory diseases, but the precise mechanisms through which hResistin's many inflammatory effects contribute to the progression of these diseases remain poorly understood. NLRP3 inflammasome activation is essential in many of these inflammatory conditions; however, there is an unmet explanation connecting hResistin with the NLRP3 inflammatory pathway. Here we describe a unique role of hResistin and its rodent homolog, resistin-like molecule alpha (RELMα) in priming and activating the NLRP3 inflammasome. Through qPCR and western blot analysis, we found that hResistin-dependent expression and secretion of high mobility group box 1 (HMGB1) in human macrophages primes the expression of NLRP3, pro-caspase-1, pro-interleukin(IL)-1β, and pro-IL-18. Co-immunoprecipitation showed that hResistin binds to Bruton's tyrosine kinase (BTK), which causes the kinase to autophosphorylate. Afterwards, BTK phosphorylates NLRP3, leading to its assembly and activation with subsequent cleavage of pro-caspase-1, pro-IL-1β, and pro-IL-18, causing initiation of the inflammasome cascade. The hResistin-dependent activation and secretion of IL-1β and IL-18 were critical to the proliferation of human pulmonary vascular smooth muscle cells. For confirmation in vivo, we studied rodent and human pulmonary hypertension (PH). Chronic hypoxia-induced PH in wild-type and RELMα KO mice showed RELMα-dependent upregulation of HMGB1, BTK, and NLRP3 in mouse lung and RELMα was linked to vascular remodeling pathways. Immunohistochemistry revealed that the majority of NLRP3-expressing cells were macrophages and the colocalization of hResistin, BTK, and NLRP3 in macrophages was increased in PH patients' lungs. Our work reveals a novel immune mechanism demonstrating hResistin is essential to the priming and activation of NLRP3. Inhibiting NLRP3 activation by blocking hResistin with a human monoclonal antibody suggests a likely therapeutic pathway for NLRP3-driven inflammatory diseases.

**Data availability statement:** All relevant data are within the manuscript and its Supporting information files.

**Funding:** This work was supported by the National Institutes of Health (NIH) Centers for Advanced Diagnostics and Experimental Therapeutics in Lung Diseases (CADET) I grant P50HL107182 (RAJ), NIH CADET II grant 5UH2HL123827-02 (RAJ), NIH grant R01HL138497 (RAJ), NIH grant R01HL166717 (RAJ, QL), and American Heart Association grant #938614/2022 (QL). These are all the funding and sources of support received during this study. None of the sponsors/ funders played a role in the study design, data collection and analysis, decision to publish, or preparation of the manuscript. There was no additional external funding received for this study.

**Competing interests:** The authors have declared that no competing interests exist.

## Introduction

Nucleotide-binding domain–like receptor protein 3 (NLRP3) is a protein complex that mediates inflammation in both infectious and sterile conditions and contributes to the progression of conditions such as diabetes, obesity, cardiovascular diseases, cancers, neuroinflammation, atherosclerosis, and other inflammation-driven diseases [1–3]. It is also strongly implicated in autoinflammatory processes [4]. The NLRP3 inflammasome is composed of three components: the sensor protein NLRP3, the adaptor protein ASC, and the effector protein caspase-1. The NLRP3 inflammasome, its components, and associated proteins are tightly regulated at two stages known as the priming stage and activation stage, but the mechanisms and regulation of these stages remain unclear [3]. The priming stage requires activation of the NF-κB pathway, which increases transcription of the NLRP3 protein and components of its downstream pathway, pro-caspase-1, pro-interleukin-1β (pro-IL-1β), and pro-interleukin-18 (pro-IL-18). The activation stage requires phosphorylation of four tyrosines in the NLRP3 protein by Bruton's tyrosine kinase (BTK) [5], leading to tertiary and quaternary changes in NLRP3 structure that enable assembly of the NLRP3 inflammasome complex and subsequent cleavage of pro-caspase-1 and cleavage and secretion of proinflammatory cytokines IL-1β and IL-18 [6,7].

We have reported previously that human resistin (hResistin) and its rodent homolog, resistin-like molecule alpha (RELMα/FIZZ1), activate damage-associated molecular pattern (DAMP) signaling and are binding partners and upstream regulators of BTK [8–12]. hResistin and RELMα are cysteine-rich secretory proteins that activate immune cells and promote the secretion of proinflammatory cytokines [13]. Similar to NLRP3, elevated levels of hResistin have been linked to diseases such as insulin resistance, atherosclerosis, diabetes, cardiac remodeling/failure, pulmonary hypertension (PH), asthma, and various autoimmune diseases [14]. Therefore, we hypothesized that hResistin/RELMα may play a critical role in both the priming (DAMP protein–dependent) and activation (BTK-dependent) of the canonical NLRP3 inflammasome pathway, mediating the pathogenesis of multiple autoinflammatory diseases (Fig 1). Moreover, we show that a human therapeutic antibody against hResistin is highly effective for reducing NLRP3 action at the site of excessive inflammation. To validate our cell-based studies in vivo and in disease, we studied these pathways in a mouse hypoxia model of PH, an NLRP3-dependent disease, and in lung tissue from patients with PH [15–19].

## Materials and methods

### NLRP3 inflammasome activation by hResistin in cultured human macrophages

Human THP-1 monocytes were purchased from ATCC (TIB-202, Manassas, VA) and were cultured in RPMI medium (ThermoFisher Scientific, R8758) with 10% heat-inactivated fetal bovine serum (FBS; Gibco, 16140−071) and 1% penicillin-streptomycin (10,000 U/mL; ThermoFisher Scientific, 15140122). THP-1 monocytes were converted to macrophages by applying phorbol myristate acetate (PMA; 50 ng/ mL, Sigma-Aldrich, P1585). Because PMA itself is involved in NLRP3 activation, we

**Graphical abstract: RESISTIN regulated priming and activation of the NLRP3 inflammasome**

**Fig 1. hResistin/RELMα regulates priming and activation of the NLRP3 inflammasome.** hResistin/RELMα activates DAMP protein HMGB1, which activates NF-κB in macrophages, leading to the priming of NLRP3 and its associated proteins. hResistin/RELMα binds and activates BTK, allowing it to activate the NLRP3 inflammasome through phosphorylation of four critical tyrosines. BTK, Bruton's tyrosine kinase; DAMP, damage-associated molecular pattern; HMGB1, high mobility group box 1; IL, interleukin; NLRP3, nucleotide-binding domain–like receptor protein 3; PH, pulmonary hypertension; PV-SMC, pulmonary vascular smooth muscle cell; RELMα, resistin-like molecule alpha. Created in BioRender. Lam, W. (2026) https://BioRender.com/1zhhut0.

washed the cells after the conversion and maintained them in a resting state as a non-biased model for NLRP3 studies, as previously described [20]. Human THP-1–derived macrophages were then treated with (1) cell culture medium only, (2) 20 nM hResistin, (3) 20 nM hResistin + 1 μg/mL high mobility group box 1 (HMGB1) Box-A antagonist, (4) 20 nM hResistin + 100 ng/mL ibrutinib, (5) 20 nM hResistin + 10 μM MCC950, (6) 20 nM hResistin + 300 ng/mL resistin antibody, (7) 20 nM hResistin + 300 ng/mL control antibody, or (8) 20 nM hResistin + NLRP3 knockout (KO) cells. LPS (100 ng/mL) and nigericin (10 μM) were used as positive control stimuli to benchmark hResistin induced responses. After 24 hours of incubation at 37°C in a 5% $CO_2$ atmosphere, culture media and whole cell lysates were collected separately. Immunoblot analysis was carried out with antibodies to BTK (Sigma-Aldrich, SAB4502936), HMGB1 (Abcam, ab79823), NLRP3 (Abcam, ab214185), pro- and cleaved caspase-1 (Abcam, ab207802), pro- and cleaved IL-18 (Invitrogen, PA5−76082), and pro- and cleaved IL-1β (Abcam, ab254360) to determine which proteins were involved in the NLRP3 activation pathway. hResistin antibodies were developed in cooperation with our commercial partners, Creative Biolabs (Shirley, NY). Lonza (London, England) and Wuxi AppTec (Cambridge, MA, and Shanghai, China) provided scale-up production. FLAG-tag recombinant human resistin protein was prepared and purified in our laboratory.

Human THP-1 cells were converted to macrophages as described above. After treating these macrophages with hResistin for 15 minutes, we assessed BTK phosphorylation by using the anti-phospho-BTK (Tyr551) antibody (Sigma-Aldrich, SAB4503802). Next, to dissect the hResistin/BTK–induced NLRP3 tyrosine phosphorylation, we treated human macrophages with 100ng/mL BTK kinase inhibitor ibrutinib (Selleckchem) before 15-minute stimulation with 200 ng/mL hResistin. Then, we analyzed for NLRP3 phospho-tyrosine (p-Y) by western blotting with anti–p-Y antibody (Sigma-Aldrich, SAB5600274).

## Human lung autopsy

We obtained autopsy lung tissue samples from 9 women (age range, 50–77 years) who had PH of varying etiologies, including systemic sclerosis (SSc)-associated pulmonary arterial hypertension, idiopathic pulmonary artery hypertension, SSc-associated pulmonary veno-occlusive disease, SSc-associated interstitial lung disease pulmonary arterial hypertension, and scleroderma-associated PH through a collaboration with the Johns Hopkins Department of Pathology (Table 1). For comparison, we also obtained biopsied lung tissue samples from control patients who had no clinical or pathologic signs of PH. This work was approved by the Johns Hopkins Human Institutional Review Board under Exemption 4 for unidentified human tissue samples.

## Mouse and in vivo treatment

Six-week-old male C57BL/6 wild-type (WT) mice (strain #000664, Jackson Laboratory) and RELMα KO mice (strain #029976, Jackson Laboratory) were maintained locally in specific pathogen-free conditions under regular hygiene monitoring. All animal experiments were used in accordance with the guidelines of the Johns Hopkins Animal Care and Use Committee. Mice were made hypoxic by exposure to 10% $O_2$ for 4 days in a Plexiglas hypoxic chamber created by Coy Labs (Ann Arbor, MI). As described previously, this custom-made hypoxia system consists of a hermetically sealed Plexiglas chamber with an oxygen-controlling system [21]. A Pro:Ox model 350 unit (Biospherix, Redfield, NY) controls and monitors the fractional concentration of $O_2$ by infusion of $N_2$ (Roberts Oxygen, Rockville, MD) balanced against an inward leak of air through holes in the chamber. A recirculating filtration system that passes through a HEPA filter, soda lime, and activated charcoal removes air contaminants and waste gases. An internal dehumidifying system maintains the chamber's humidity between 30% and 50%. Normoxic control mice were exposed to room air in a Plexiglas box in standard mouse

**Table 1. Hemodynamic data and diagnostic markers of patients with pulmonary hypertension.**

| Patient | Age (years) | Sex | Diagnosis | mPAP (mmHg) | CO/CI L/min/ L/min/m² | PAWP mmHg | PVR WU |
|---|---|---|---|---|---|---|---|
| 1 | 50 | F | SSc-PAH | 57 | 3/1.51 | 14 | 14 |
| 2 | 65 | F | IPAH | 65 | 4.7/2.9 | 7 | 12 |
| 3 | 67 | F | SSc-PAH | 48 | 4.3/2.1 | 10 | 8.9 |
| 4 | 58 | F | SSc-PVOD | 47 | 8.7/4 | 6 | 4.7 |
| 5 | 70 | F | SSc-PAH | 43 | 2.3/1.3 | 21 | 9.6 |
| 6 | 74 | F | SSc-PAH/PVOD | 38 | 4.8 | 8 | 6.25 |
| 7 | 73 | F | SSc-PAH | 60 | 3.27/1.69 | 12 | 14.7 |
| 8 | 77 | F | SSc-PAH | 33 | 4.2/2.6 | 10 | 5.5 |
| 9 | 53 | F | SSc-ILD-PH | 40 | 6.5/3.5 | 13 | 6.2 |

CI, cardiac index; CO, cardiac output; F, female; IPAH, idiopathic pulmonary artery hypertension; mPAP, mean pulmonary arterial pressure; PAWP, pulmonary arterial wedge pressure; PVR, pulmonary vascular resistance; SSc-ILD-PH, systemic sclerosis-associated interstitial lung disease; SSc-PAH, systemic sclerosis-associated pulmonary arterial hypertension; SSc-PVOD, systemic sclerosis-associated pulmonary veno-occlusive disease; WU, wood units.

cages. To further interrogate the downstream NLRP3 pathways, we treated mice with (1) hResistin/RELMα antibody (Lonza) or (2) control IgG1. The hResistin/RELMα antibody and control IgG1 were administered systemically at 4 mg/kg with an intraperitoneal injection twice a week beginning 1 week before hypoxia. After 4 days of continuous hypoxic conditions, all mice were euthanized via CO2 overexposure for 5–10 minutes. When no signs of life were detected (no heartbeat and respiration), cervical dislocation was performed as trained, according to the guidelines of the Johns Hopkins Animal Care and Use Committee. In total, 48 mice were used for our studies and all efforts to minimize suffering and distress were taken into consideration. Although no mice met the criteria, if any had experienced a weight loss of ≥15%, they would have been euthanized as our humane endpoint. Expression and localization of the RELMα-BTK-NLRP3 signaling molecules were determined by qPCR, western blotting, and tissue immunostaining in the lung tissues collected.

## Quantitative reverse-transcriptase PCR

Human THP-1 cells were converted to macrophages as described previously. They were then treated with (1) cell culture medium only, (2) 20 nM hResistin, (3) 20 nM hResistin + 1 µg/mL high mobility group box 1 (HMGB1) Box-A antagonist, (4) 20 nM hResistin + 100 ng/mL ibrutinib, (5) 20 nM hResistin + 10 µM MCC950, (6) 20 nM hResistin + 300 ng/mL resistin antibody, or (7) 20 nM hResistin + 300 ng/mL control antibody. After 6, 12, 18, or 24 hours of treatment, total RNA was isolated with the RNeasy kit (Qiagen, 74004) according to the manufacturer's protocol, and then 500 ng was reverse transcribed into cDNA. Quantitative PCR was carried out on an ABI 7500 fast real-time PCR system (Applied Biosystems) using primers from Integrated DNA Technologies. Fold changes in gene expression were acquired by the delta method and normalization to beta actin. Total RNA from mouse lung tissues were isolated, reverse transcribed, and analyzed following the same protocol as described above. The primers used in this process were for HMGB1, IL-1β, IL-18, NLRP3, BTK, Caspase-1, human resistin, mouse RELMα, and β-actin (S1 Table).

## Western blot analysis

Mouse lungs were collected in RIPA buffer (Sigma) supplemented with 1 mM phenylmethylsulfonyl fluoride (PMSF, Thermo Scientific, 36978), 1 mM $Na_4VO_3$, and protease inhibitor mixture (Roche, 116974980011). The lungs were lysed with homogenization beads (0.9–2.0 mm; SSB14B, Bullet Blender) in a Bullet Blender at 4°C, vortexed, and then centrifuged. Cell pellets were suspended in cell lysis buffer (Cell Signaling Technology, 9803) supplemented with 1 mM PMSF, 1 mM $Na_4VO_3$, and protease inhibitor mixture (Roche, 116974980011), lysed by sonication, and then centrifuged. The concentration of isolated protein (lung tissues or cell pellets) was measured by the bicinchoninic acid method (BCA kit, Bio-Rad). The supernatants were mixed in sodium dodecyl sulfate (SDS) 5× sample loading buffer (NuPAGE, Invitrogen) at 99°C for 10 minutes and then subjected to SDS-polyacrylamide gel electrophoresis (PAGE). After electrophoresis, proteins were transferred onto polyvinylidene difluoride membranes and immunoblotted with antibodies to RELMα (R&D Systems, AF1523), β-actin (Cell Signaling Technology, 3700), BTK (Sigma-Aldrich, SAB4502936), HMGB1 (Abcam, ab79823), NLRP3 (Abcam, ab214185), pro- and cleaved caspase-1 (Abcam, ab207802), pro- and cleaved IL-18 (Invitrogen, PA5−6082), and pro- and cleaved IL-1β (Abcam, ab254360) overnight at 4°C and then probed with horseradish peroxidase–conjugated secondary antibodies (Bio-Rad) for 2 hours. Protein bands were visualized by chemiluminescence (ECL, Amersham Pharmacia Biotech).

To determine whether hResistin activates AKT and ERK1/2 phosphorylation in human pulmonary vascular smooth muscle cells (PVSMCs) through mature IL-1β and IL-18, we used conditioned media from hResistin-treated macrophages to treat PVSMCs and assessed phosphorylated AKT and ERK1/2 levels by immunoblotting. Human PVSMCs were serum-starved with 0.5% FBS for 24 hours and then grown in one of the following conditions: (1) cell culture medium only, (2) macrophage-conditioned medium treated with 20 nM hResistin, (3) macrophage-conditioned medium treated with 20 nM hResistin + 300 ng/mL resistin antibody (4) macrophage-conditioned medium treated with 20 nM hResistin + 300 ng/mL resistin antibody, (5) macrophage-conditioned medium with 120 ng/mL IL-1β antibody, (6) macrophage-conditioned medium with 120 ng/

mL IL-18 antibody, (7) starved medium (0.5% FBS) + 5 ng/mL IL-1β protein, or (8) starved medium + 5 ng/mL IL-18 protein. After 30 minutes, cells were collected for western blot analysis and immunoblotting with p-AKT (Cell Signaling Technology, 4060), t-AKT (Cell Signaling Technology, 9272), p-ERK1/2 (Cell Signaling Technology, 4370), and total ERK1/2 (Cell Signaling Technology, 4695) antibodies. Protein bands were visualized by chemiluminescence (ECL, Amersham Pharmacia Biotech).

## ELISA

Supernatant concentrations of IL-1β and IL-18 levels were measured by commercially available ELISA kits (RayBiotech, Cat# ELH-IL1b-2, Cat# ELH-IL18–2). The optical densities of the tested samples were compared with the values obtained from serial dilution of the respective recombinant cytokine in each kit. Each sample was tested in duplicate.

## Immunohistochemistry staining

We used immunofluorescence staining to detect the expression of NLRP3, Mac2, myeloperoxidase (MPO), CD79b, BTK, and hResistin/RELMα in lung tissue sections of human autopsy samples and WT and KO mice. After deparaffinization of tissue, rehydration, and antigen retrieval, sections were treated with anti-NLRP3 (Abcam, ab214185) and anti-Mac2 (Cedarlane, CL8942LE), anti-MPO (R&D Systems, AF3667), anti-CD79b (Abcam, ab134147), anti-BTK (Sigma-Aldrich, SAB4502936), anti-hResistin (R&D Systems, AF1359), or anti-RELMα (R&D Systems, MAB1523) antibodies overnight at 4°C and then with Alexa Fluor 488-donkey anti-mouse IgG (Jackson ImmunoResearch, 715-545-150) and Cy3-donkey anti-rabbit IgG (Jackson ImmunoResearch, 711-166-152) for double fluorescence staining. Sections were then incubated with the appropriate fluorochrome-coupled secondary antibody (Jackson ImmunoResearch). Finally, all sections were mounted in ProLong® Gold anti-fade reagent with DAPI (Invitrogen). Staining was imaged and tissue sections were analyzed by confocal microscopy (Leica SPE DMI8).

## Co-immunoprecipitation assay

We assessed binding of hResistin to human BTK by immunoprecipitation. We prepared 75 μg of anti-FLAG antibody for coupling to the resin. FLAG-hResistin was mixed with hypoxic macrophage cell lysate in buffer containing 30 mM HEPES, 150 mM NaCl, and 2 mM β-mercaptoethanol at pH 7.5 and incubated for 30 minutes at 30°C. The mixture was further incubated overnight at 4°C with AminoLink Plus Coupling Resin and washed twice with 500 μL of the same buffer; protein was eluted according to the manufacturer's instructions (ThermoScientific, 26149). We loaded 30% and 70% of the sample as input and elution fractions, respectively, and analyzed protein binding by SDS-PAGE and immunoblotting using monoclonal anti-FLAG M2-peroxidase (HRP) or anti-BTK antibody (Sigma-Aldrich, SAB4503802).

## Caspase-1 activation assay

Caspase-1 activity in hResistin-treated THP-1–converted macrophages was determined by detecting the cleavage of YVAD-AFC using the Abcam Caspase-1 Assay Kit (Fluorometric, ab39412). Briefly, hResistin-treated THP-1–converted macrophages were treated with (1) cell culture medium only, (2) 100 ng/mL ibrutinib, (3) 20 nM hResistin, (4) 20 nM hResistin + 100 ng/mL ibrutinib, (5) 20 nM hResistin + 300 ng/mL resistin antibody. After 24 hours of incubation at 37°C in a 5% $CO_2$ atmosphere, cell pellets were collected, homogenized, and centrifuged. Then, the supernatants were incubated with YVAD-AFC (1 mM) at 37°C for 2 hours. Fluorescence was measured at the excitation/emission wavelengths of 400/505 nm according to the manufacturer's instructions.

## BrdU assay

Primary human pulmonary vascular artery smooth muscle cells [from Lonza (Morristown, NJ); 2,000 cells/well in 96-well plates] were cultured in Smooth Muscle Cell Growth Medium SmGM™-2 BulletKit™ (Lonza, CC-3182) with FBS + penicillin-streptomycin, serum-starved for 24 hours, and then stimulated with (1) cell culture medium only,

(2) medium + 0.5% FBS, (3) medium + 5% FBS, (4) medium + 0.5% FBS + 20nM lab-made recombinant hResistin, (5) medium + 3 µg lab-made recombinant hResistin, (6) 10ng/mL platelet-derived growth factor, (7) non-stimulated macrophage-conditioned medium, (8) conditioned medium from macrophages treated with 20nM hResistin for 30 minutes, (9) macrophage-conditioned medium treated with 20 nM hResistin + 300 ng/mL resistin antibody, (10) macrophage-conditioned medium with 120 ng/mL IL-1β antibody, or (11) macrophage-conditioned medium with 120 ng/mL IL-18 antibody. After 30 minutes, the medium was removed and cells were incubated for 48 hours with 200 µL of smooth muscle cell basal media + 0.5% FBS at 37°C. Then, 5-bromo-2′-deoxyuridine (BrdU) was added for 24 hours to label proliferating cells. The next day, BrdU ELISA was performed according to the manufacturer's instructions (Roche, 11647229001) to measure and quantify cell proliferation.

## Statistical analysis

Data are presented as the mean ± SEM. Comparisons between two groups were analyzed by Student's *t* test, and comparisons of multiple groups were analyzed by one-way ANOVA followed by the Newman-Keuls post hoc test, unless otherwise noted. The Tukey post hoc test was used for immunoblotting analysis. All statistical analyses were carried out with Prism 7.0e (GraphPad Software, La Jolla, CA). A *p* value <0.05 was considered statistically significant.

## Results

### hResistin-dependent DAMP activation primes the NLRP3 inflammasome

Because NLRP3 inflammasome activation requires priming before activation, we first assessed the involvement of hResistin in the priming stage. Given that hResistin stimulates the expression and secretion of the DAMP protein HMGB1, which can initiate priming, we hypothesized that hResistin regulates the priming stage through HMGB1 [8,10]. Therefore, we treated THP-1 cells that were differentiated into macrophage-like cells with hResistin and then blocked HMGB1 with Box-A inhibitor. Similarly, we used hResistin antibody or control antibody to bind and block hResistin. The control antibody was an inactive isotype of the hResistin antibody.

After 24 hours, western blot analysis showed that hResistin increased the protein expression level of NLRP3, as well as pro-caspase-1, pro-IL-1β, and pro-IL-18 (Fig 2A–C). These elevations in protein expression were comparable to THP-1 derived macrophages treated with 100ng/mL LPS and 10µM of nigericin (S1A, B Fig). Both hResistin treated cells and LPS and nigericin treated cells showed similar increases in total NLRP3, pro-caspase-1, pro-IL-1β, and pro-IL-18 levels. Additionally, the mRNA levels for these molecules also increased significantly, with fold change averages ranging from 6 to 75 by 24 hours (S2 Fig). Interestingly, this activity was blocked in samples treated with HMGB1 Box-A inhibitor or hResistin antibody, indicating that both HMGB1 and hResistin are linked regulators of the priming stage. Blocking hResistin reduced the protein expression and fold change levels of HMGB1 in the THP-1 cells, reducing the downstream expression of NLRP3. To determine the involvement of BTK and NLRP3 in the priming stage, we used ibrutinib (a BTK inhibitor) and MCC950 (an NLRP3 inhibitor). In addition to blocking NLRP3, we used NLRP3 KO THP-1–differentiated macrophages to further confirm our hypothesis. However, compared with hResistin-treated samples, we found no significant differences in ibrutinib- or MCC950-treated samples or in NLRP3 KO cells, indicating that blocking BTK-dependent NLRP3 activation does not affect the priming stage. These data indicate that hResistin-dependent DAMP activation is essential for the priming stage of NLRP3 inflammasome activation.

### hResistin binds to and initiates autophosphorylation of BTK in human macrophages

As we and others have shown, BTK requires autophosphorylation before it can phosphorylate other proteins [12,22–24]. We hypothesized that hResistin regulates NLRP3 activation by inducing the autophosphorylation of BTK. After THP-1–converted macrophages were exposed to hypoxia for 24 hours, hResistin expression increased nearly 40-fold compared with that in normoxic macrophages (S3A Fig). Stimulation of macrophages with hResistin dose-dependently increased the

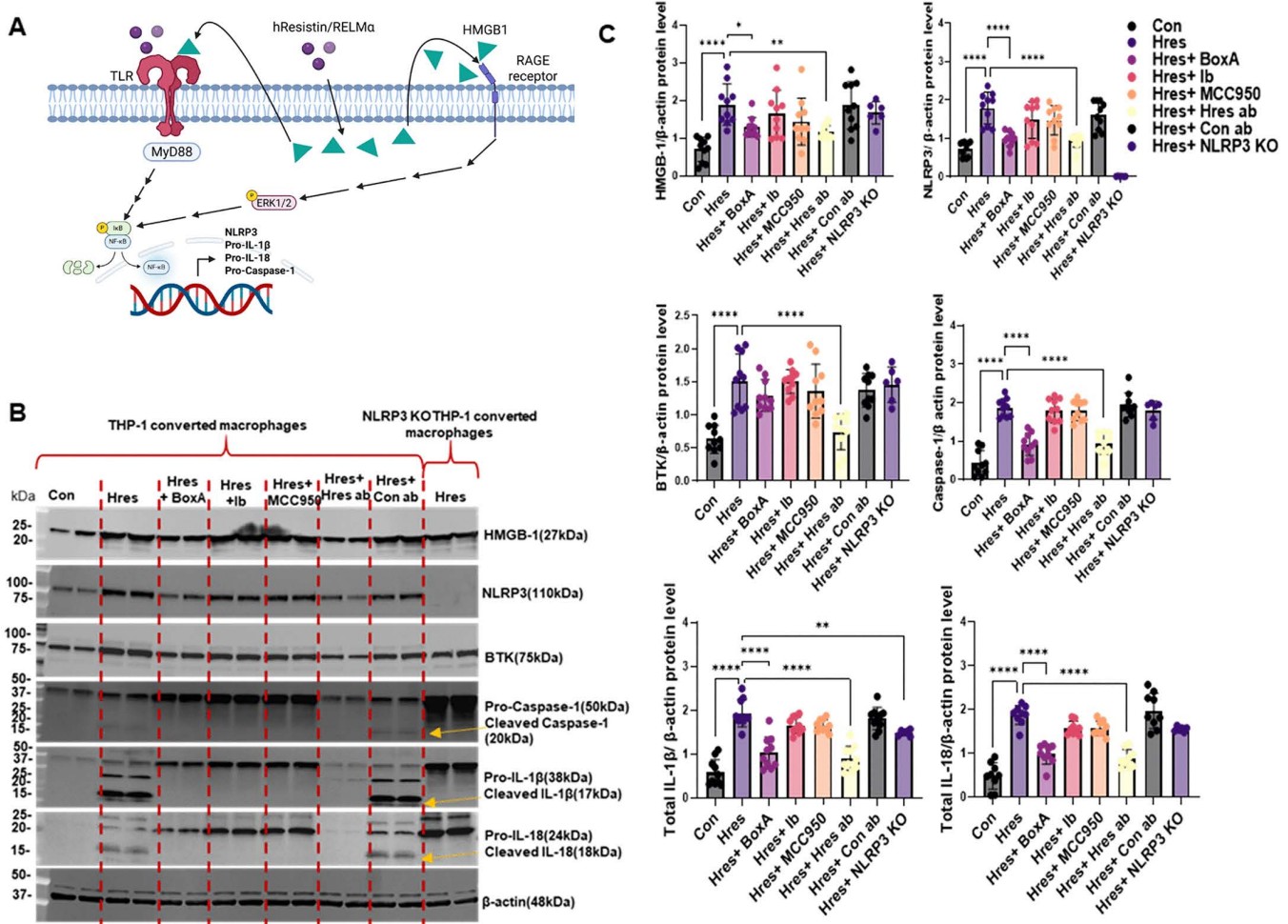

**Fig 2. hResistin upregulates the expression of BTK, HMGB1, NLRP3, pro-IL-1β, pro-IL-18, and pro-caspase-1 and the cleavage of pro-IL-1β, pro-IL-18, and pro-caspase-1 in human macrophages.** Human THP-1–derived macrophages were cultured under the conditions shown for 24 hours. (**A**) Schematic diagram of proposed pathway. (**B**) Western blot images of HMGB1, NLRP3, BTK, pro- and cleaved caspase-1, pro- and cleaved IL-1β, and pro- and cleaved IL-18. (**C**) Quantitative analysis of each protein from panel B. Data represent means±SD (n=10 per group except the NLRP3 KO group, where n=6). *$p < 0.05$, **$p < 0.01$, ****$p < 0.0001$. ab, antibody; Box-A, HMGB1 inhibitor; BTK, Bruton's tyrosine kinase; Con, control; Ib, ibrutinib; HMGB1, high mobility group box 1; Hres, human resistin; IL, interleukin; KO, knockout; NLRP3, nucleotide-binding domain–like receptor protein 3; RAGE, receptor for advanced glycation end products. Created in BioRender. Lam, W. (2026) https://BioRender.com/1zhhut0.

protein level of BTK while increasing BTK mRNA levels roughly 60-fold (S2 and S3B-C Figs). We have previously reported that mouse RELMα is a binding partner of mouse BTK [12], and because RELMα is a mouse homolog of hResistin, we tested whether human BTK binds hResistin using pulldown assays as previously described [5]. Indeed, we found that BTK pulled down hResistin (S3D Fig). Moreover, hResistin stimulation increased phospho-BTK, an effect that was blocked by hResistin blocking antibody, indicating that hResistin is involved in BTK autophosphorylation (S3E, F Fig).

## hResistin-BTK signaling stimulates NLRP3 phosphorylation leading to its full activation and production of active caspase-1, IL-1β, and IL-18

After inflammasome priming, the next activation step induces formation of the NLRP3 inflammasome complex, which consists of active NLRP3, ASC, and active capsase-1 (Fig 3A). Active caspase-1 cleaves pro-IL-1β and pro-IL-18 to their

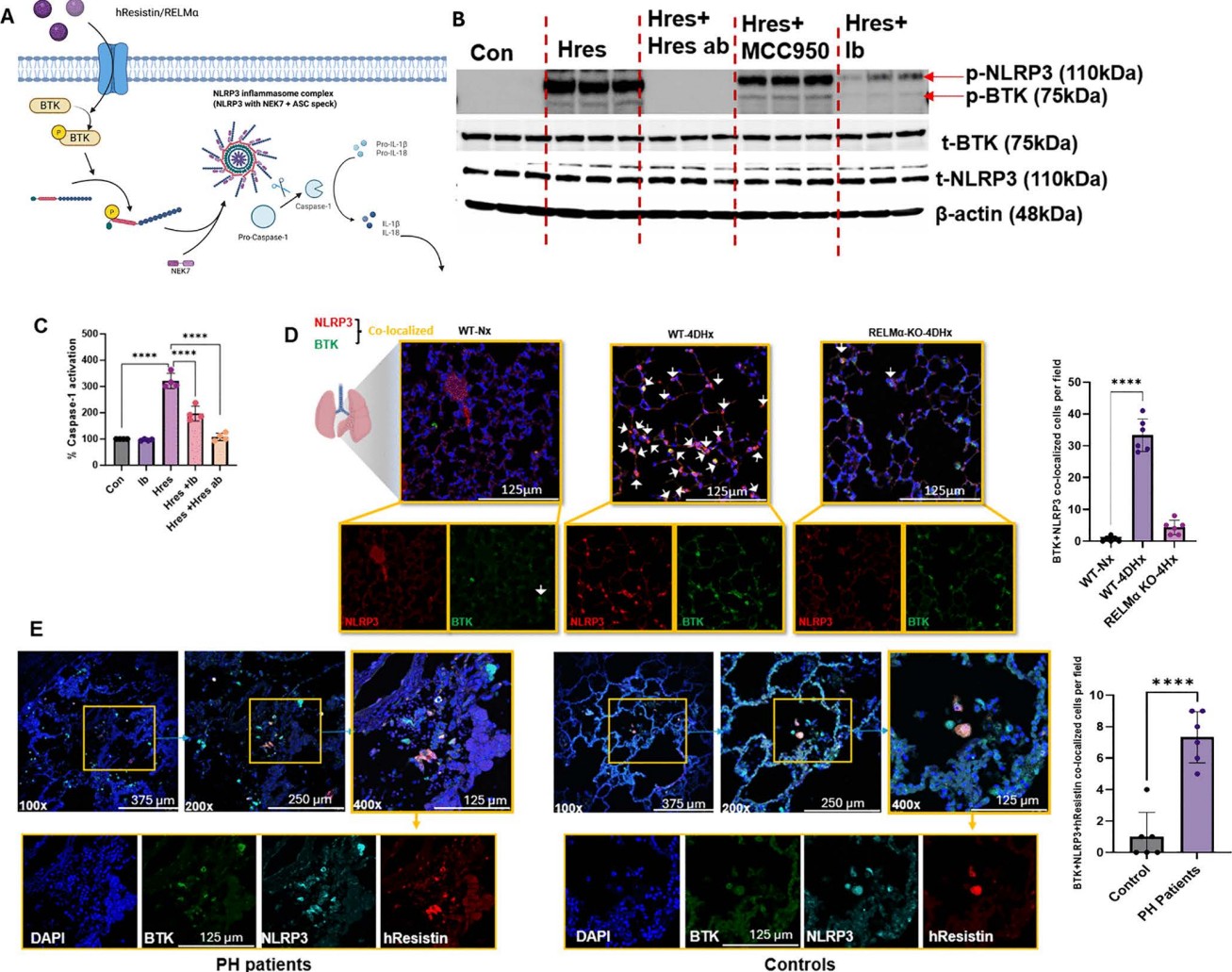

**Fig 3. hResistin stimulates NLRP3 inflammasome activation in human macrophages.** (**A**) Schematic presentation of proposed pathway. (**B**) Human macrophages were cultured in 6-well plates under the conditions shown for 15 minutes before being washed and lysed for western blot analysis with antibodies to phospho-tyrosine (p-Y) and NLRP3 (image is representative of 6 repeats). (**C**) Human macrophages were cultured under the conditions shown for 12 hours. PBS was used as a control. Cells were washed, lysed, and assayed for their ability to cleave a fluorescent caspase-1 substrate, YVAD-AFC. Values were normalized to PBS controls. All conditions were run in duplicate wells, and two independent experiments were performed. Error bars represent the mean ± SD (n = 4). ****$p < 0.001$ versus PBS. (**D**) Hypoxia upregulates BTK and NLRP3 colocalization in C57BL/6 WT mice but not in RELMα KO mice. Immunofluorescence images of NLRP3 and BTK in lung tissues of mice kept under normoxic or hypoxic conditions for 4 days. Lung sections were stained with anti-NLRP3 (red) and BTK (green). The arrowheads point to cells positively stained for BTK and NLRP3 (yellow). The upper images are shown at higher magnification (400×); the lower panels display separate channels. Representative photograph of n = 6 mice per group. (**E**) Human resistin colocalized with BTK and NLRP3 in patients with PH. Immunofluorescence images of lung tissue slices from PH patients. Sections were stained with anti-hResistin (red) and co-stained with anti-BTK (green) and anti-NLRP3 (cyan) antibodies. The arrowheads point to cells positively stained for hResistin, BTK, and NLRP3. Separate channels are displayed in the lower panels. Original magnification: 100 × , 200 × , and 400 × . 4DHx, 4-day hypoxic; ab, antibody; BTK, Bruton's tyrosine kinase; Con, control; Hres, human resistin; Ib, ibrutinib; KO, knockout; NLRP3, nucleotide-binding domain–like receptor protein 3; Nx, normoxic; PBS, phosphate-buffered saline; PH, pulmonary hypertension; RELMα, resistin-like molecule alpha; WT, wild-type. Created in BioRender. Lam, W. (2026) https://BioRender.com/1zhhut0.

active forms, resulting in their secretion [3]. Phosphorylation of four tyrosines in NLRP3 by BTK is a key step in NLRP3 activation [5,6]. Because hResistin was shown to be an upstream activator of BTK, we hypothesized that hResistin-dependent BTK activation is critical to NLRP3 phosphorylation and its activation. We treated differentiated THP-1 cells with hResistin for 24 hours with or without hResistin antibody, ibrutinib, or MCC950 and immunoblotted with phospho-tyrosine antibodies to assess changes in phosphorylation status of NLRP3 [5]. hResistin-treated samples showed two bands corresponding to the molecular weights of phosphorylated NLRP3 and phosphorylated BTK (Fig 3B). Treatment with ibrutinib completely blocked formation of both bands, confirming the upstream role of BTK on NLRP3 phosphory-lation. Immunoblots of hResistin antibody–treated samples did not show phospho-BTK or phospho-NLRP3 bands, con-firming that hResistin contributes to NLRP3 activation via autophosphorylation of BTK. We further verified this finding by using a caspase-1 activation assay. As expected, blocking with either ibrutinib or hResistin antibody reduced caspase-1 activation significantly (Fig 3C). Examining the secretion of IL-1β and IL-18 in the supernatant, we found that hResistin significantly increased the levels of both cytokines (S4 Fig). Just as using either ibrutinib or hResistin antibody reduced the activation of caspase-1, using these antibodies also reduced the amount of IL-1β and IL-18 in the supernatant. Since the use of MCC950 and NLRP3 KO cells reduced their levels as well, the NLRP3 inflammasome appears to be a critical component in hResistin-induced secretion of these cytokines.

To test the role of mouse RELMα in inflammasome activation in an in vivo system, we used lung sections from 4-day hypoxic WT and RELMα KO mice and performed immunohistochemistry. Similar to previous findings, NLRP3 and BTK colocalization was significantly higher in the lungs of hypoxic WT mice than in the lungs of normoxic mice (Fig 3D). How-ever, hypoxia did not increase levels of BTK and NLRP3 colocalization in RELMα KO mice, supporting our hypothesis that hResistin/RELMα is critical to the activation of NLRP3 through BTK phosphorylation. Similarly, levels of BTK, NLRP3, and hResistin were higher in the lungs of PH patients than in those of healthy controls, with hResistin colocalizing with both NLRP3 and BTK (Fig 3E).

## RELMα upregulates proinflammatory phenotypes in hypoxic mouse lungs

Next, we tested our hypothesis in WT and RELMα KO mice with active NLRP3. We have previously shown that a 4-day hypoxic exposure increases lung RELMα expression in mice and initiates the process of vascular remodeling [25,26]. Here, the WT mice were kept for 4 days under normoxic and hypoxic conditions with or without hResistin antibody or control antibody. NLRP3 and IL-1β were significantly increased by hypoxia, but this increase was attenuated in the group treated with hResistin antibody (Fig 4A). Next, both WT and RELMα KO mice were exposed to hypoxia for 4 days. Immu-noblot analysis of isolated lungs revealed that HMGB1, BTK, and NLRP3 were significantly increased in hypoxic WT mice compared with levels in normoxic WT mice (Fig 4B–D). Remarkably, hypoxia did not increase HMGB1, BTK, or NLRP3 in RELMα KO mice, indicating the importance of RELMα to activation of these proteins in vivo.

## Macrophages are the main source of hResistin and NLRP3 in the lungs of hypoxic mice and patients with PH

Because elevated levels of hResistin have been linked to PH, we used PH as a model to study hResistin regulation of the NLRP3 inflammasome. Indeed, inflammasomes are currently being evaluated as a novel therapeutic target for PH [19]. To determine which type of immune cells in the lung express the highest level of NLRP3, we collected lung sections from mice after 4 days of hypoxia and from patients with PH. We carried out immunohistochemistry staining for NLRP3 and MAC2 (macrophage), myeloperoxidase (neutrophils), and CD79b (B cell) markers. Almost all macrophages in mouse lungs expressed NLRP3, and nearly 80% of NLPR3-expressing cells in human lungs were macrophages (Fig 5A). Con-sistent with previous literature, macrophages were found to be the main source of hResistin in humans (Fig 5B). Colocal-ization of NLRP3, BTK, and hResistin in macrophages confirmed macrophages to be the main source of hResistin and NLRP3 in lung tissue of PH patients (Fig 5B and C). Our mouse model of PH showed increased colocalization of RELMα

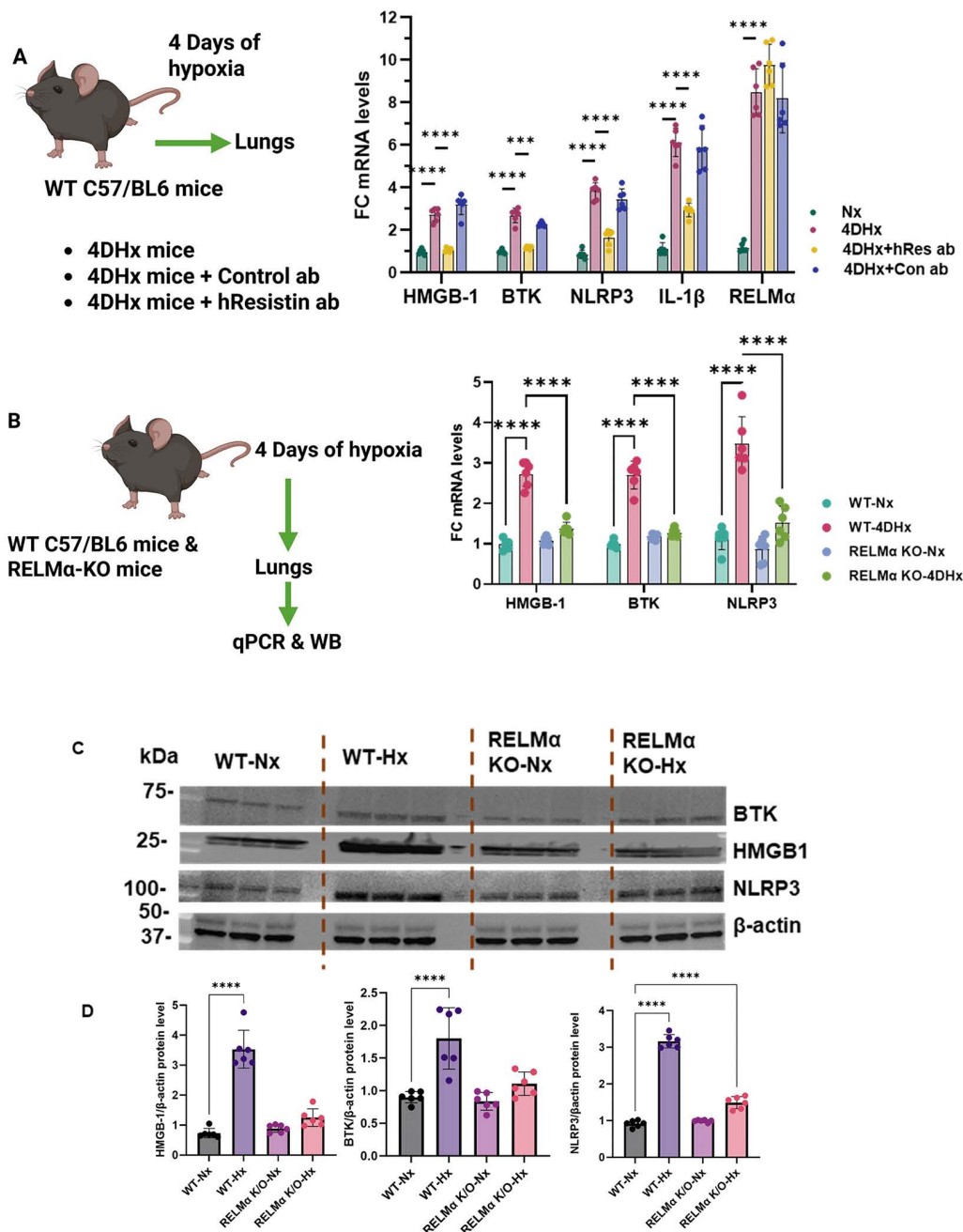

**Fig 4. RELMα upregulates proinflammatory phenotypes in hypoxic mouse lungs. (A)** Quantitative RT-PCR analysis of HMGB1, BTK, NLRP3, pro-IL-1β, and RELMα gene expression in lung tissues collected from WT C57BL/6 mice that were kept for 4 days under hypoxic conditions with or without hResistin antibody or control antibody. Data represent means±SD (n=6 per group). **(B)** Quantitative RT-PCR analysis of HMGB1, BTK, and NLRP3 gene expression in WT and RELMα KO mouse lungs after exposure to normoxic and hypoxic conditions (n=6 per group). **(C)** Western blot images show protein expression of BTK, HMGB1, and NLRP3 in lungs from WT and RELMα KO mice after exposure to normoxic and hypoxic conditions. **(D)** Quantitative analysis of data in panel **C.** Data represent means±SD (n=6 per group). ***p<0.001, ****p<0.0001. 4DHx, 4-day hypoxic; BTK, Bruton's tyrosine kinase; FC, fold change; HMGB1, high mobility group box 1; hResistin, human resistin; KO, knockout; NLRP3, nucleotide-binding domain–like receptor protein 3; Nx, normoxic; RELM, resistin-like molecule; WB, western blot; WT, wild-type. Created in BioRender. Lam, **W.** (2026) https://BioRender.com/1zhhut0.

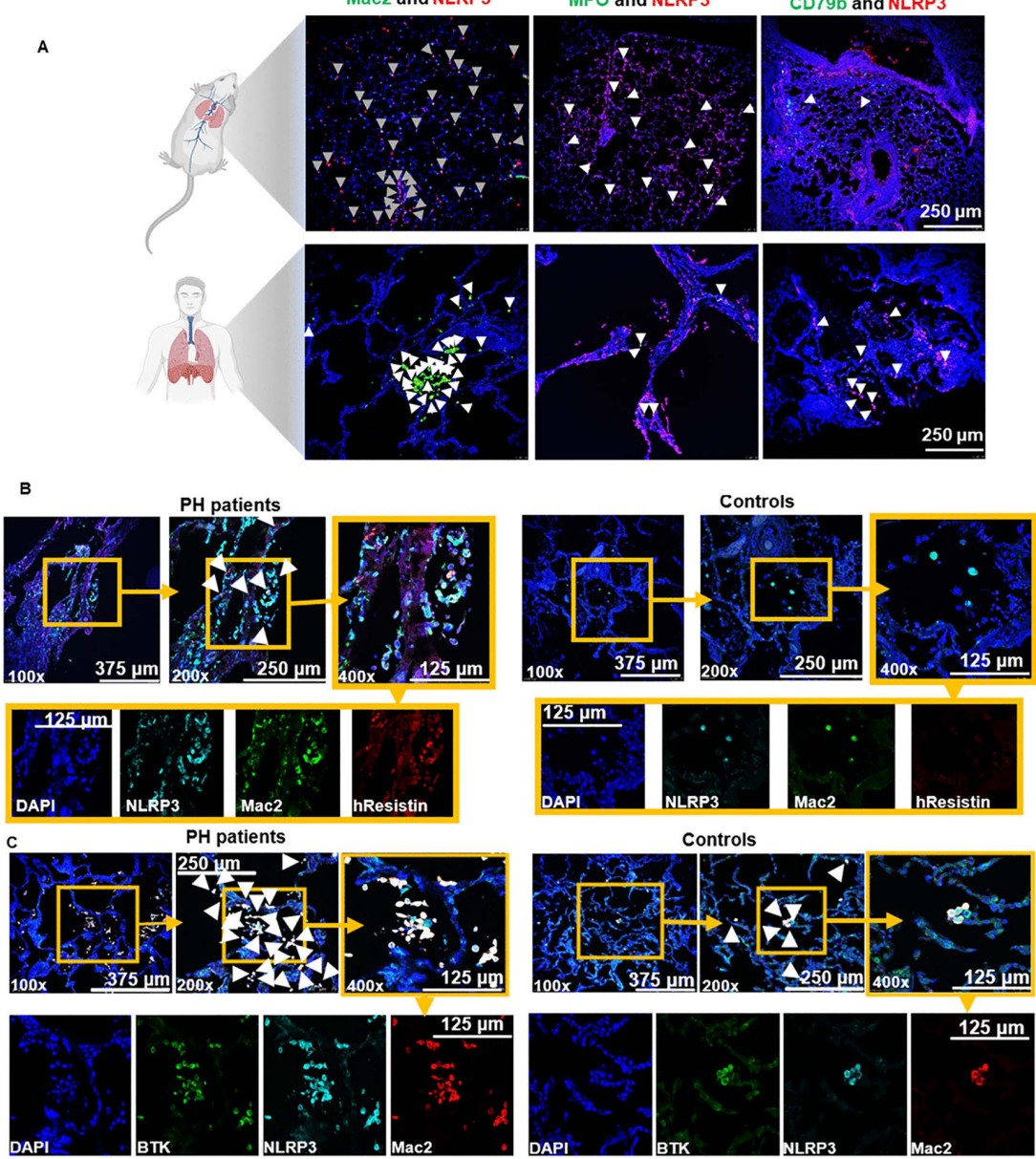

**Fig 5. Macrophages are the main source of NLRP3 in hypoxic mouse lungs and PH patient lungs. (A)** Immunofluorescence images of NLRP3 and immune cell markers in lung tissues of 4-day hypoxic mice and lung sections from PH patients. Lung sections were stained with anti-NLRP3 (red) and/or Mac2 (green), MPO (green), or CD79b (green). The images are shown at 100×magnification. Representative photograph of 6 mice per group and 3 human lung samples. **(B)** Immunofluorescence images of lung tissue slices from PH patients show hResistin colocalization with NLRP3 in macrophages. Sections were stained with anti-human resistin (red) and co-stained with anti-Mac2 (green) and anti-NLRP3 (cyan) antibodies. The arrowheads point to cells positively stained for hResistin, Mac2, and NLRP3. Separate channels are displayed in the lower panels. Original magnification: 100×, 200×, and 400×. **(C)** Immunofluorescence images show colocalization of NLRP3 and BTK in macrophages. Sections were stained with anti-human BTK (green) and co-stained with anti-Mac2 (red) and anti-NLRP3 (cyan) antibodies. The arrowheads point to cells positively stained for human macrophages, BTK, and NLRP3. Separate channels are displayed in the lower panels. Original magnification: 400×. BTK, Bruton's tyrosine kinase; hResistin, human resistin; MPO, myeloperoxidase; NLRP3, nucleotide-binding domain–like receptor protein 3; PH, pulmonary hypertension. Created in BioRender. Lam, **W.** (2026) https://BioRender.com/1zhhut0.

with BTK as well when the mice were exposed to hypoxia for 4 days (S5 Fig). This colocalization is consistent with our past findings of RELMα being a binding partner of BTK [12].

**hResistin promotes human PVSMC proliferation through macrophage-derived mature IL-1β and IL-18**

The data thus far indicate that hResistin initiates early immune responses through priming and activation of the NLRP3 inflammasome. However, a knowledge gap remains in regard to how this early activation might heighten the pathogenesis of PH. Hence, we explored how hResistin-BTK-NLRP3 signaling leads to the development of PVSMC proliferation. We hypothesized that hResistin stimulates AKT and ERK1/2 phosphorylation in human PVSMCs through mature IL-1β and IL-18. Therefore, we treated human PVSMCs with hResistin-treated macrophage-conditioned medium and measured phospho-AKT and phospho-ERK1/2 levels by immunoblot analysis. The macrophage-conditioned medium did increase levels of phospho-AKT and phospho-ERK1/2, but this activity was blocked by conditioned medium containing hResistin antibody or MCC950. Further, the addition of either IL-1β blocking antibody or IL-18 blocking antibody significantly decreased the phosphorylation. Treatment of PVSMCs with mature IL-1β and IL-18 proteins upregulated the phosphorylation of AKT and ERK1/2 (Fig 6A and B), confirming that both proteins directly affect phosphorylation.

We have previously found that high concentrations of hResistin can stimulate human PVSMC proliferation [10]. Therefore, we tested the effects of high and low concentrations of hResistin on human PVSMC proliferation in a BrdU assay, with platelet-derived growth factor as a positive control (Fig 6C). Because the conditioned media contains 5% FBS, we used 5% FBS to assess baseline cell proliferation. Similar to previous results, hResistin-treated conditioned media significantly increased PVSMC proliferation. Cell proliferation was significantly downregulated in samples treated with hResistin antibody, IL-1β blocking antibody, or IL-18 blocking antibody, further supporting the premise that hResistin-regulated, macrophage-derived mature IL-1β and IL-18 mediate post-injury proliferative responses in the lungs. To further confirm the effect of hResistin on PVSMC proliferation, we analyzed levels of matrix metalloproteinase 1 (MMP-1) by western blot. MMP-1 was induced by hResistin-treated conditioned media and significantly reduced by blocking IL-1β or IL-18 (Fig 6D and E).

## Discussion

Inflammasomes are key components of macrophage-mediated immunity [19,27]. Additionally, hResistin/RELMα is considered a primary marker of Th2 inflammation [21,28–30]. We now prove a critical interaction of hResistin/RELMα is in the priming and activation of the NLRP3 inflammasome, suggesting a novel pathway and target for the wide-ranging autoinflammatory diseases mediated by NLRP3. We found that hResistin/RELMα utilizes the HMGB1 and NF-κB priming pathway to increase both the mRNA and protein levels of NLRP3 and the pro- forms of caspase-1, IL-1β, and IL-18. Additionally, we found that hResistin/RELMα binds and activates BTK, allowing it to auto phosphorylate. Phosphorylated BTK is then able to phosphorylate four tyrosines on NLRP3, leading to the activation of the NLRP3 inflammasome, allowing the cleavage of the pro-forms of caspase-1, IL-1β, and IL-18. Then the mature IL-1β and IL-18 can be secreted from macrophages. Our findings represent a novel understanding of hResistin/RELMα's role in regulating the NLRP3 inflammasome in the macrophage, as well as in inflammatory diseases and autoinflammation, presenting a novel target for therapy.

Collective evidence in the published literature has shown early NLRP3 inflammasome activation in macrophages to be the key mechanism of inflammation-driven, macrophage-induced inflammatory diseases [18,31–34]. Because macrophages are the main cellular source of hResistin and NLRP3, we focused on macrophages to study the critical regulatory role of hResistin/RELMα [35–37]. After macrophages were stimulated with hResistin, we observed increased NLRP3 and the pro- forms of caspase-1, IL-1β, and IL-18 at both the mRNA and protein levels. We previously reported that hResistin, an upstream regulator of HMGB1 [10], activates NF-κB through HMGB1. Inhibiting HMGB1 significantly downregulated the expression levels of NLRP3 and other NLRP3 components, supporting our hypothesis that HMGB1 is the intermediate mediator of hResistin/RELMα in the priming pathway. Acetylated HMGB1 secreted from cells activates NF-κB through

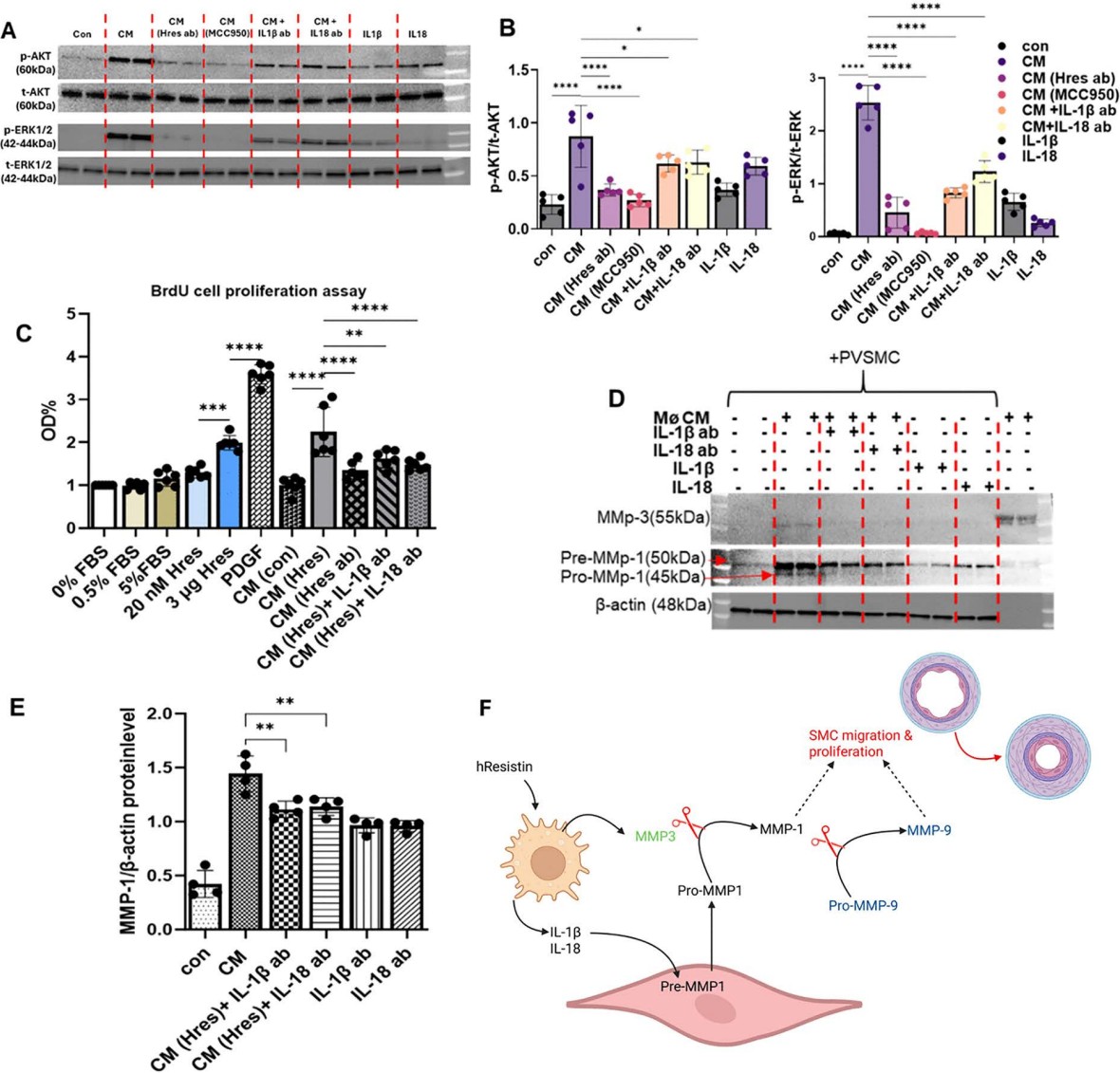

**Fig 6. hResistin induces macrophage-derived mature IL-1β and IL-18 to promote human PVSMC proliferation. (A)** Images of western blots for p-AKT, t-AKT, p-ERK1/2, and t-ERK1/2 protein levels in human PVSMCs that were serum-starved for 24 hours and then cultured for 30 minutes in (1) medium only; conditioned medium from macrophages treated with (2) 20 nM hResistin, (3) 20 nM hResistin + 300 ng/mL resistin antibody, (4) 20 nM hResistin + 10 μM (4.04 μg/mL) MCC950, (5) 120 ng/mL IL-1β antibody, (6) or 120 ng/mL IL-18 antibody; (7) starved media (0.5% FBS) + 5 ng/mL IL-1β protein; or (8) starved media + 5 ng/mL IL-18 protein. **(B)** Quantitative analysis of data in panel A. Data represent means ± SD (n = 5 per group). **(C)** Induction of PVSMC proliferation by IL-1β and IL-18 derived from hResistin-treated macrophages. Data represent means ± SD (n = 6 per group). **(D)** hResistin regulates MMP-driven PVSMC proliferation through IL-1β and IL-18 derived from hResistin-treated macrophages. Human PVSMCs were serum-starved for 24 hours and then cultured for 24 hours with (1) medium only; conditioned media from macrophages treated with (2) 20 nM hResistin, (3) 120 ng/mL IL-1β antibody, or (4) 120 ng/mL IL-18 antibody; (5) starved media + 5 ng/mL IL-1β protein; (6) starved media + 5 ng/mL IL-18 protein; or (7) conditioned media alone. **(E)** Quantitative analysis of data in panel D. Data represent means ± SD (n = 4 per group). **(F)** Schematic presentation of the proposed pathway of SMC migration and proliferation. $*p < 0.05$, $**p < 0.01$, $***p < 0.001$, $****p < 0.0001$. ab, antibody; CM, conditioned medium; con, control; FBS, fetal bovine serum; Hres; human resistin; IL, interleukin; NS-CM, non-stimulated conditioned media; MMP, matrix metalloproteinase; OD, optical density; p-, phosphorylated; PDGF, platelet-derived growth factor; PVSMC, pulmonary vascular smooth muscle cell; t-, total. Created in BioRender. Lam, W. (2026) https://BioRender.com/1zhhut0.

Toll-like receptor 4 (TLR4) or receptor for advanced glycation end products (RAGE) receptors [11,38,39]. Although previous literature has suggested a role for HMGB1 in the priming step of inflammasome activation, it has not been clearly demonstrated in detail [9]. Our data regarding hResistin regulation of the inflammasome explains our early observation that NF-κB activates IL-1β, IL-6, and TNFα in macrophages [40–43]. Our data strongly support a mechanism whereby hResistin–HMGB1 signaling promotes NLRP3 inflammasome priming in macrophages (Fig 1).

After the production of inflammasome components in the priming step, the inflammasome must be activated in a second step [3]. Phosphorylated BTK induces the phosphorylation of four tyrosines in NLRP3 [5]; this phosphorylated NLRP3 facilitates the subsequent subcellular relocalization and oligomerization of NLRP3, ASC polymerization, and full NLRP3 assembly, which ultimately leads to the cleavage of pro-caspase-1 and cleavage and secretion of IL-1β and IL-18 [5–7]. We have reported previously that RELMα is a binding partner and activator of BTK [12]. RELMα induces BTK phosphorylation to promote the migration of myeloid cells [12]. More importantly, we showed that binding of hResistin causes conformational changes of BTK that facilitate its autophosphorylation and allow presentation of phospho-BTK to NLRP3 tyrosine phosphorylation sites. Treatment of human macrophages with the BTK kinase inhibitor ibrutinib demonstrated that hResistin/BTK signaling is required for downstream NLRP3 inflammasome complex assembly, for cleavage of pro-caspase-1, and for cleavage and secretion of IL-1β and IL-18. Caspase-1 activation was absent in both NLRP3 KO and MCC950-treated cells, further confirming that hResistin induced caspase-1 activation is NLRP3-dependent. Similar to the priming pathway, we showed that activating hResistin/RELMα–BTK signaling is essential to regulate NLRP3 tyrosine phosphorylation, leading to assembly and full activation of the NLRP3 inflammasome in macrophages and IL-1β/IL-18 secretion from these cells.

In this study, we demonstrated that macrophages were activated by hResistin/RELMα to secrete active IL-1β and IL-18, which initiate early immune responses, with IL-1β potentially acting as an endogenous cytokine that can prime the NLRP3 inflammasome through the IL-1β receptor in other macrophages [3]. We propose that priming and activation of the NLRP3 inflammasome provides a potential mechanism through which hResistin mediates inflammation in multiple diseases. Previously, we reported that RELMα is markedly upregulated (64-fold) in rodent models of PH and that hResistin levels are high in the lungs of patients with PH [11,44]. Moreover, hResistin expression level correlated with the severity of PH in humans and predicted mortality [45,46]. To assess the in vivo role of hResistin/RELMα in human disease, we characterized hResistin/RELMα activation of the inflammasome pathway in rodent and human models of PH, an inflammation-driven disease whose pathophysiology is clearly linked to hResistin and RELMα [25,26,43,44,47–49]. We found that hResistin both primes and activates the NLRP3 inflammasome through HMGB1 and BTK, respectively, to initiate the inflammation that stimulates and possibly maintains vascular remodeling in PH. By regulating the NLRP3 inflammasome, hResistin induces phosphorylation of AKT and ERK1/2 in PVSMCs, which are common proliferation markers in lung vascular remodeling. hResistin blocking antibody significantly reduced this phosphorylation, indicating the direct involvement of hResistin. We further confirmed this finding by showing that hResistin conditioned media induced human PVSMC proliferation in a BrdU cell-proliferation assay.

It is still unclear how the diverse inflammatory effects of hResistin/RELMα are integrated or how they amplify and sustain inflammation during disease progression. hResistin/RELMα activates Th2 differentiation of macrophages, and Th2 stimuli also can induce RELMα expression through IL-4, IL-13, and STAT-6, creating an amplification loop [21,25,26,43]. hResistin/RELMα can initiate inflammatory responses by activating DAMP pathway proteins [8,10,11]. Thus, hResistin/RELMα signaling may act as a crucial hub of a positive-feedback loop to trigger, amplify, and sustain inflammation through its immunoregulatory activities in disease progression. We hypothesized that for this hResistin-BTK-NLRP3 pathway to have a continuing long-term effect on autoinflammation, it could have a connection with adaptive immunity and crosstalk between activated macrophages with other immune cells. It may also reflect the hResistin-driven pathways for transition from innate to adaptive inflammatory responses to injury in the lung and merits further investigation.

Our findings show that hResistin is a major regulator and effector of macrophages and a critical driver of a proinflammatory phenotype. This study addresses critical gaps in understanding the primary mechanism by which the NLRP3

inflammasome is primed and activated in macrophage/immune cells and defines a new mechanism of hResistin/RELMα actions in inflammatory disease. Our work demonstrates that our specific human monoclonal therapeutic antibody against hResistin is highly effective and highlights hResistin as a novel target to reduce NLRP3 action at the site of excessive inflammation. Our findings shed light on hResistin-driven immune responses and suggest a novel immunotherapeutic approach for a variety of autoinflammatory disorders, including PH.

## Supporting information

**S1 Fig. hResistin increases the protein expression of NLRP3, pro-caspase-1, IL-1β, and IL-18 to levels similar to LPS and nigericin treated macrophages.** Human THP-1-derived macrophages were cultured under the conditions shown for 24 hours. (**A**) Western blot analysis of total NLRP3, pro-caspase-1, pro- and cleaved IL-1β, and pro- and cleaved IL-18. (**B**) Quantitative analysis of data in panel A. Data represents means ± SEM (n = 6). ****$p < 0.0001$; ns, not significant. Hres, human resistin; IL, interleukin; LPS, lipopolysaccharide; NLRP3, nucleotide-binding domain-like receptor protein 3.
(TIF)

**S2 Fig. hResistin upregulates the gene expression of HMGB1, BTK, NLRP3, pro-IL-1β, pro-IL-18, and pro-caspase-1 in human macrophages.** Human THP-1–derived macrophages were cultured under the conditions shown for 24 hours. Quantitative PCR analysis of HMGB1, BTK, NLRP3, pro-caspase-1, pro-IL-1β, and pro-IL-18 shows relative mRNA levels. Data represent means ± SD (n = 5 per group). ****$p < 0.0001$. ab, antibody; Box-A, HMGB1 inhibitor; BTK, Bruton's tyrosine kinase; Con, control; FC, fold change; Ib, Ibrutinib; HMGB1, high mobility group box 1; Hres, human resistin; IL, interleukin; KO, knockout; NLRP3, nucleotide-binding domain–like receptor protein 3.
(TIF)

**S3 Fig. hResistin binding to BTK upregulates BTK autophosphorylation in human macrophages.** (**A**) Quantitative RT-PCR analysis of hResistin gene expression in THP-1 differentiated macrophages exposed to normoxia or hypoxia for 24 hours. Data represent means ± SD (n = 6 wells per group). (**B**) Western blot analysis of total BTK expression in human macrophages after exposure to hResistin for 24 hours upregulates BTK expression. (**C**) Quantitative analysis of data in B. hResistin dose-dependently increased BTK in human macrophages. Data represent means ± SEM (n = 6). (**D**) THP-1 macrophages were exposed to hypoxia for 24 hours. Co-immunoprecipitation assays were carried out with anti-FLAG antibody–coupled resin. FLAG-hResistin proteins with macrophage cell lysates were used as inputs. Input: before passing through the anti-FLAG antibody–embedded resin; E1: first elution; E2: second elution; positive control (+): recombinant BTK and hResistin proteins. (**E**) Human THP-1–derived macrophages were treated as shown before western blot analysis. (**F**) Quantitative analysis of each protein from panel E. Data represent means ± SD (n = 6 per group). ****$p < 0.0001$. ab, antibody; BTK, Bruton's tyrosine kinase; Con, control; FC, fold change; Hres, human resistin; Hx, hypoxic; IP, immuno-precipitated; Nx, normoxic; p-, phosphorylated; t-, total.
(TIF)

**S4 Fig. hResistin increases the secretion of IL-1β and IL-18 in human macrophages.** Human THP-1–derived macrophages were cultured under the conditions shown for 24 hours. Supernatants of each sample were used according to the manufacturer's protocols to quantify the respective cytokine concentrations. Data represents means ± SEM (n = 6). ****$p < 0.0001$. ab, antibody; Box-A, HMGB1 inhibitor; BTK, Bruton's tyrosine kinase; Con, control; Ib, Ibrutinib; HMGB1, high mobility group box 1; Hres, human resistin; IL, interleukin; KO, knockout; LPS, lipopolysaccharide; NLRP3, nucleotide-binding domain–like receptor protein 3.
(TIF)

**S5 Fig. RELMα colocalizes with BTK in murine hypoxic lung.** Paraffin-embedded lung sections were dual stained with a rabbit anti-RELMα polyclonal antibody that was visualized by an Alexa Fluor 488-conjugated goat anti-rabbit IgG

antibody and a mouse anti-BTK monoclonal antibody that was visualized by a Cy3-conjugated donkey anti-mouse IgG antibody.
(TIF)

**S1 Table. Primers used for quantitative RT- PCR.**
(TIF)

**S2 Table. Reagents and materials used for this study.**
(DOCX)

**S1 Raw Images. Raw western blot images used for figures.**
(PDF)

## Acknowledgments

We would like to acknowledge Dr. Marc Halushka from the Department of Pathology, Johns Hopkins University School of Medicine, Baltimore, MD, USA, for providing autopsy lung sections.

## Author contributions

**Conceptualization:** Udeshika Kariyawasam, John Skinner, Andrea Cox, Qing Lin, Roger A. Johns.

**Data curation:** Udeshika Kariyawasam, Winson Lam, John Skinner.

**Formal analysis:** Udeshika Kariyawasam, Winson Lam.

**Funding acquisition:** Qing Lin, Roger A. Johns.

**Investigation:** Udeshika Kariyawasam, Winson Lam, John Skinner, Rituparna Chakrabarti, Andrea Cox, Paul M. Hassoun, Qing Lin.

**Methodology:** Udeshika Kariyawasam, John Skinner.

**Project administration:** Roger A. Johns.

**Supervision:** Roger A. Johns.

**Visualization:** Udeshika Kariyawasam, Winson Lam.

**Writing – original draft:** Udeshika Kariyawasam, Roger A. Johns.

**Writing – review & editing:** Udeshika Kariyawasam, Winson Lam, Rituparna Chakrabarti, Andrea Cox, Paul M. Hassoun, Roger A. Johns.

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
