## [Decision Letter · Decision Letter 0]

11 Dec 2025

Dear Dr. Johns,

Thank you for submitting your manuscript to PLOS ONE. After careful consideration, we feel that it has merit but does not fully meet PLOS ONE’s publication criteria as it currently stands. Therefore, we invite you to submit a revised version of the manuscript that addresses the points raised during the review process.

We look forward to receiving your revised manuscript.

Kind regards,

Kota V Ramana, Ph.D.

Academic Editor

PLOS One

Journal Requirements:

[This work was supported by National Institutes of Health (NIH) Centers for Advanced Diagnostics and Experimental Therapeutics in Lung Diseases (CADET) I grant P50HL107182 (RAJ), NIH CADET II grant 5UH2HL123827-02 (RAJ), NIH grant R01HL138497 (RAJ), NIH grant R01HL166717 (RAJ, QL), and American Heart Association grant #938614/2022 (QL).

https://www.nhlbi.nih.gov/science/lung-diseases

https://www.nih.gov/about-nih/nih-almanac/national-heart-lung-blood-institute-nhlbi

https://www.heart.org/

The sponsors/funders did not play any role in the study design, data collection and analysis, decision to publish, or preparation of the manuscript.].

5. We note that Figure 5 in your submission may contain copyrighted images. All PLOS content is published under the Creative Commons Attribution License (CC BY 4.0), which means that the manuscript, images, and Supporting Information files will be freely available online, and any third party is permitted to access, download, copy, distribute, and use these materials in any way, even commercially, with proper attribution. For more information, see our copyright guidelines: http://journals.plos.org/plosone/s/licenses-and-copyright.

1. You may seek permission from the original copyright holder of Figure 5 to publish the content specifically under the CC BY 4.0 license.

Reviewers' comments:

Reviewer's Responses to Questions

**Comments to the Author**

1. Is the manuscript technically sound, and do the data support the conclusions?

Reviewer #1: Yes

Reviewer #2: Yes

2. Has the statistical analysis been performed appropriately and rigorously?

Reviewer #1: Yes

Reviewer #2: No

3. Have the authors made all data underlying the findings in their manuscript fully available?

Reviewer #1: Yes

Reviewer #2: Yes

4. Is the manuscript presented in an intelligible fashion and written in standard English?

Reviewer #1: Yes

Reviewer #2: No

Reviewer #1: Dear Dr. Roger A Johns.

I would like to thank you for the opportunity to review this excellent, meticulous, and well-planned work. The results are described meticulously and elegantly.

My only concern is regarding the manipulation of the western blot images. Unfortunately, the resolution of images 2, 3, 4, and 6 in the manuscript is not detailed enough to allow for a decent review.

However, in image S2, panels B, D, and E, which show western blot images, there are some concerning features, such as anomalous uniformity in backgrounds and load controls, as well as possible excessive background correction and assembly of independent gels.

Is it possible to obtain the original unedited files (raw tiff files for each membrane), full exposure (“full uncut blots”), and acquisition metadata from you?

This would give me peace of mind regarding such beautiful research work.

Reviewer #2: The manuscript by Kariyawasam et al. investigates human resistin (hResistin) and its rodent homolog, resistin-like molecule alpha (RELMα), as upstream regulators of NLRP3 inflammasome priming and activation in macrophages. They act via HMGB1–NF-κB–dependent transcriptional priming and BTK-mediated phosphorylation pathways, respectively. The study demonstrates that hResistin induces HMGB1 release to prime NLRP3 and its downstream pro-cytokines, engages BTK to promote NLRP3 tyrosine phosphorylation, caspase 1 activation, and maturation of IL 1β and IL 18, and drives PVSMC proliferation through these cytokines, with upregulation of this axis confirmed in hypoxic mouse lungs and pulmonary hypertension patient tissues, and loss of hypoxia-induced signalling in RELM-deficient mice. The following major and minor issues should be resolved before the manuscript is suitable for publication.

Major comments

1. qRT PCR data are presented at 24 h and described as assessing pro IL 1β, pro caspase 1, and pro IL 18 “forms,” but mRNA measurements reflect gene expression and cannot distinguish zymogen versus cleaved protein species; only immunoblotting and ELISA can discriminate inactive precursors from active cytokines, and gene expression kinetics should be evaluated at earlier time points distinct from protein analyses to avoid misinterpretation and overstatement throughout the text.

2. The study does not include a canonical positive control for NLRP3 inflammasome activation, such as LPS priming followed by ATP or nigericin stimulation, which is necessary to benchmark hResistin/RELMα-induced responses against established NLRP3 activation paradigms.

3. The authors state that both canonical and non-canonical inflammasome pathways and their downstream signalling are examined, but experimental evidence distinguishing these pathways is not clearly defined or systematically presented.

4. The graphical abstract depicts multiple cytokines, yet the manuscript does not comprehensively measure or report several of these mediators, rendering the schematic potentially misleading and requiring alignment of depicted factors with experimentally quantified analytes.

5. The overall design links in vitro macrophage signalling to in vivo pulmonary hypertension models; however, the 4-day hypoxia protocol in mice likely captures early inflammatory and remodelling events rather than fully established chronic pulmonary hypertension, which limits direct translational extrapolation to long-standing human disease.

6. Phospho tyrosine immunoblotting is interpreted as specific evidence for BTK- and NLRP3-directed phosphorylation but lacks site specificity; BrdU incorporation assays quantify PVSMC proliferation but do not assess migration despite claims involving MMP-1-driven motility; and no cytokine ELISA data are provided to quantify IL-1β and IL-18 secretion into supernatants.

7. The experimental framework appropriately incorporates pharmacological (ibrutinib, MCC950) and genetic (NLRP3 knockout, RELM knockout) controls, and the use of PMA differentiated, rested THP 1 macrophages reduces PMA related bias; nonetheless, the co immunoprecipitation strategy relies on FLAG tagged recombinant hResistin incubated with lysates, which may generate non physiological interactions and should be complemented by approaches that detect endogenous complexes in situ (e.g., proximity ligation assays or native co IP).

8. The 4-day hypoxia model robustly induces inflammatory signalling but does not establish full pulmonary hypertension physiology, as right ventricular systolic pressure, right ventricular hypertrophy, and fibrosis endpoints are not reported, while autopsy lung samples from PH patients represent heterogeneous, end-stage disease with non standardized treatment and comorbidity histories.

9. The data are used to argue that hResistin is an upstream therapeutic target for NLRP3 inhibition via monoclonal antibody blockade, but phospho tyrosine blots are overinterpreted as mapping to specific BTK and NLRP3 residues without corroborating evidence from site-directed mutagenesis or mutational loss-of-function studies.

10. The main conceptual innovation is the integration of hResistin/RELMα into both the priming and activation phases of NLRP3 signaling, extending prior work on BTK–NLRP3 regulation; however, mechanistic depth is limited regarding HMGB1–NF κB–RAGE/TLR4 circuitry, ASC speck formation, and upstream NF κB activation dynamics in this context.

11. Important gaps remain, including the absence of direct assays of NLRP3 oligomerisation and ASC polymerisation (e.g., ASC cross-linking or speck imaging) and the lack of controls for alternative BTK activators (such as TLR ligands), which may confound attribution of BTK activation solely to hResistin/RELMα.

12. Although RELM knockout abolishes hypoxia-induced HMGB1, BTK, and NLRP3 upregulation in mouse lungs, alternative explanations persist because HMGB1 can signal through multiple receptors, including TLR4 and RAGE, which are not dissected experimentally in this study.

13. The proposed hResistin–BTK–NLRP3 axis would be substantially strengthened by validation in primary human macrophages, by genetic silencing (siRNA/shRNA) of hResistin and BTK, and by visualisation of NLRP3–ASC specks to demonstrate bona fide inflammasome assembly.

Minor comments

1. Nomenclature and labelling should be standardised: for example, “hResistin” is used in the text, whereas figure panels and the graphical abstract sometimes show “Hresistin”; a single, consistent designation should be applied throughout.

2. Several textual elements require correction and harmonisation, including “pro caspase 1” to “pro caspase 1” in the Abstract and consistent ibrutinib dosing units in the Methods section (currently reported as 100 ng/mL and 60 μM).

3. Key references need to be specified: the BTK–NLRP3 tyrosine phosphorylation literature (currently cited broadly as references 5–7) should clearly identify the modified residues, and primers listed in Supplementary Table 1 should be quantitatively integrated into the main Results (e.g., fold change ranges and statistics).

4. Several figure-related issues require clarification: Fig. 2C is annotated as n=9, but bar appearance suggests fewer replicates; Fig. 3B phospho tyrosine bands are not annotated with molecular weights; and Fig. 6D refers to “MMP-driven migration,” although only MMP 1 protein levels are shown without accompanying migration assays.

5. Abbreviations and numerical entries are sometimes inconsistent or contain typographical errors, including variable use of “pro IL 1” versus “pro interleukin 1” and apparent formatting issues in Table 1 hemodynamic data (e.g., “31.51” for cardiac output/cardiac index).

6. Supplemental Figures 1–3 are cited but not fully described in the main text, and some immunohistochemistry panels lack explicit scale bars, particularly in Figure 5, which should be corrected for interpretability.

7. Specific English and grammar revisions are recommended, for example:

o “hResistins many inflammatory effects contribute to the progression of these diseases remain poorly understood” should be revised for clarity (e.g., insertion of appropriate punctuation and verb agreement).

o “pro caspase 1, pro interleukin 1, and pro IL 18” should be presented consistently with standard cytokine nomenclature.

o “causes the kinase to autophosphorylate, allowing BTK to phosphorylate NLRP3, leading to its assembly...” could be streamlined to avoid repetition and clarify sequence.

o “was linked to vascular remodelling pathways” should be rewritten to specify the type of linkage and supporting data.

8. Some supplemental figures lack error bars, and there are discrepancies between n values stated in the main figures (e.g., Fig. 2C, n=9) and those in corresponding supplemental data (e.g., Supplementary Fig. 1, n=5), which should be reconciled.

9. Figure formatting should be globally harmonised, including consistent representation of fold change (use either “FC” or “Fc” uniformly), ordering and labelling of panels, and alignment of panel titles and legends to reduce confusion for readers.

**Do you want your identity to be public for this peer review?** For information about this choice, including consent withdrawal, please see our For information about this choice, including consent withdrawal, please see our Privacy Policy .

Reviewer #1: **Yes:** Juan AlpucheJuan Alpuche

Reviewer #2: **Yes:** Vivek Phani VarmaVivek Phani Varma

---

## [Author Response · Author response to Decision Letter 1]

24 Jan 2026

Thank you all for your reviews. We have addressed all comments and added new data to our manuscript. The file "Response to Reviewers" has all the comments.

---

## [Decision Letter · Decision Letter 1]

17 Mar 2026

Human resistin is critical to activation of the NLRP3 inflammasome in macrophages

PONE-D-25-60813R1

Dear Dr. Johns,

We’re pleased to inform you that your manuscript has been judged scientifically suitable for publication and will be formally accepted for publication once it meets all outstanding technical requirements.

Kind regards,

Kota V Ramana, Ph.D.

Academic Editor

PLOS One

Additional Editor Comments (optional):

Reviewers' comments:

Reviewer's Responses to Questions

**Comments to the Author**

Reviewer #1: All comments have been addressed

Reviewer #2: All comments have been addressed

2. Is the manuscript technically sound, and do the data support the conclusions?

Reviewer #1: Yes

Reviewer #2: Yes

3. Has the statistical analysis been performed appropriately and rigorously?

Reviewer #1: Yes

Reviewer #2: Yes

4. Have the authors made all data underlying the findings in their manuscript fully available?

Reviewer #1: Yes

Reviewer #2: Yes

5. Is the manuscript presented in an intelligible fashion and written in standard English?

Reviewer #1: Yes

Reviewer #2: Yes

Reviewer #1: Thank you for addressing all the commentaries. I have no additional concerns.

Reviewer #2: The authors have satisfactorily addressed all the criteria and adequately responded to the queries raised during the previous submission. I am satisfied with their explanations and have no further comments on the revised manuscript.

**Do you want your identity to be public for this peer review?** For information about this choice, including consent withdrawal, please see our For information about this choice, including consent withdrawal, please see our Privacy Policy .

Reviewer #1: **Yes:** Juan AlpucheJuan Alpuche

Reviewer #2: **Yes:** Vivek Phani Varma DVivek Phani Varma D

---

## [Editor Report · Acceptance letter]

PONE-D-25-60813R1

PLOS One

Dear Dr. Johns,

I'm pleased to inform you that your manuscript has been deemed suitable for publication in PLOS One. Congratulations! Your manuscript is now being handed over to our production team.

Kind regards,

on behalf of

Dr. Kota V Ramana

Academic Editor

PLOS One